# FGF7 as an essential mediator for the onset of ankylosing enthesitis related to psoriatic dermatitis

Shin Ebihara[1] , Yuji Owada[2], Masao Ono[1,3]

**IL-17A plays an important role in the pathology of psoriasis and psoriatic arthritis (PsA). However, the pathogenic association between the skin and joint manifestations in PsA is not completely understood. In this study, we initially observed that IL-17A and FGF7 induced endochondral ossification in the mouse entheseal histoculture. Importantly, the responses of endochondral ossification by IL-17A stimulation were strongly inhibited by the treatment of a blocking antibody to FGF receptor 2IIIb, which is the receptor of FGF7, suggesting that FGF7 acts as a downstream factor of IL-17A in the endochondral ossification in the culture. Next, using the animal PsA model, the administration of an anti-FGF receptor 2IIIb antibody resulted in significant suppression of ankylosing enthesitis but not dermatitis. Collectively, our findings indicate that augmented IL-17A in PsA dermatitis induces the elevation of FGF7 levels in joint enthesis and results in a non-redundant role of FGF7 signaling in the development of ankylosing enthesitis in PsA.**

## Introduction

Psoriatic arthritis (PsA) is a chronic inflammatory arthritis, a type of spondyloarthritis (SpA), associated with psoriasis (Gladman et al, 2005). The characteristic feature of PsA is new bone formation, including heterotopic endochondral ossification, which leads to joint ankylosis (Rahimi & Ritchlin, 2012). The prevalence of PsA in the psoriasis population varies widely, most recently estimated to range from 5 to 40% (Christophers et al, 2010). Previous studies have demonstrated higher prevalence of PsA in more severe (Tey et al, 2010; Eder et al, 2012) or longer lasting (Christophers et al, 2010; Eder et al, 2012) psoriasis. Thus, skin disease is considered as a main risk factor for co-occurrence of joint disease (Gisondi et al, 2022). Clinical therapeutics for both psoriasis and PsA are commonly considered and used (Stiff et al, 2018). Although psoriatic skin lesions and joints in PsA share common genetic antecedents and a number of similar mechanisms at the cellular and molecular levels

(Ruiz et al, 2012; Stuart et al, 2015), the pathogenic connection remains unclear (Belasco & Wei, 2019; Winthrop et al, 2019).

Many studies have indicated that IL-17A is involved in PsA pathogenesis in humans. Higher levels of IL-17A have been detected in the whole blood (Gado et al, 2021) and synovial fluid (Leipe et al, 2010) of patients with PsA. Increased levels of infiltration of IL-17A–producing cells, including CD4+ T cells (Leipe et al, 2010), CD8+ T cells (Menon et al, 2014), and γδ T cells (Guggino et al, 2016), have been observed in the joints of patients with PsA. It has been reported that the levels of IL-17A in whole blood (Gado et al, 2021) and IL-17A+ CD8+ T cells in joints (Menon et al, 2014) are significantly correlated with the disease activity of PsA. Recently, inverse-variance weighted analysis with single-nucleotide polymorphisms indicated a causal association between increased IL-17A levels and the risk of PsA (Wu et al, 2021). In a phase III clinical study, treatment with secukinumab (formerly AIN457, Novartis), an anti-IL-17A monoclonal antibody, was associated with significant improvements in the clinical symptoms of both skin and joint of PsA as compared to placebo (McInnes et al, 2015). Furthermore, ixekizumab (formerly LY2439821, Eli Lilly), another IL-17A inhibitor, is also effective for the treatment of PsA (Mease et al, 2017). Thus, IL-17A plays a critical role as a causal factor in pathogenesis of PsA. However, the detailed role of IL-17A in joint pathogenesis in patients with PsA remains unknown (McGonagle et al, 2019; Zwicky et al, 2020).

FGF/FGF receptor (FGFR) signaling plays a central role in physiological endochondral ossification and bone homeostasis (Zhang et al, 2023). Among them, *FGFR2* has been identified as a significant epigenomic gene by a genome-wide DNA methylation study in the cartilage of osteoarthritis (OA) (Jeffries et al, 2016), in which endochondral ossification is a fundamental pathway in the progression (Pulsatelli et al, 2013). *FGFR2* is expressed at higher levels in OA cartilage than in normal cartilage and involved in accelerated chondrocyte differentiation (Jeffries et al, 2016). Accumulating evidence has shown that a gain-of-function mutation in *FGFR2* causes several human bone developmental diseases including Crouzon syndrome (Jabs et al, 1994; Wilkie et al, 1995), Jackson–Weiss syndrome (Jabs et al, 1994), and Apert syndrome (Wilkie et al, 1995). FGF7, also known as a keratinocyte growth factor, is a paracrine and/or autocrine mediator in bone formation (Zhang

---

[1]Department of Pathology, Tohoku University Graduate School of Medicine, Sendai, Japan  [2]Department of Organ Anatomy, Tohoku University Graduate School of Medicine, Sendai, Japan  [3]Department of Clinical Laboratory, National Hospital Organization Mito Medical Center, Ibaraki-machi, Japan

Correspondence: ebihara@med.tohoku.ac.jp

https://doi.org/10.26508/lsa.202403073 vol 8 | no 4 | e202403073 **1 of 15**

et al, 2023). FGF7 binds to its high-affinity receptor FGFR2IIIb (also known as KGFR), which is a splice variant of *FGFR2*. A previous study demonstrated that FGF7 is expressed at significantly higher levels in OA meniscal cells than in normal meniscal cells (Sun et al, 2010). Moreover, FGF7 has been shown to play a positive role in new bone formation by local delivery of FGF7 in the mandible defects of rats (Poudel et al, 2017). In the tracheal cartilage wound healing model, it has been shown that FGF7 expresses in both chondrocytes and perichondrial cells and that FGFR2IIIb expression is limited to proliferating chondrocytes in response to injury (Abo et al, 2010).

Spontaneously occurring arthropathy in aged male DBA/1J mice is characterized not only by transient acute inflammation surrounding the enthesis in the peripheral joint, including the fingers, but also by heterotopic endochondral ossification, which leads to ankylosis (Lories et al, 2004). Our previous study showed that these mice spontaneously and readily develop psoriasis-like dermatitis when the mice caged together, and, importantly, an initial symptom of ankylosing enthesitis chronologically onsets after dermatitis (Ebihara et al, 2015). In this model, neutralizing IL-17A inhibited the development of dermatitis and ankylosing enthesitis (Ebihara et al, 2015).

Here, we investigated the role of IL-17A in endochondral bone formation. To prove a possible association between IL-17A and FGF7, we examined an effect of a blocking antibody against FGFR2IIIb, a receptor for FGF7, on disease manifestations in vivo, as well as a change of indicators for inflammation and endochondral ossification in our histoculture system. We demonstrated that IL-17A acts as an upstream mediator of FGF7 and that FGF7 plays an essential role in co-occurrence of cutaneous and joint diseases in PsA.

# Results

### Induction of the endochondral ossification by IL-17A stimulation in entheseal histoculture

To investigate the pathogenesis of entheseal lesions, such as endochondral ossification in PsA, we developed entheseal histoculture as a research tool. To explore the effect of IL-17A on endochondral ossification, we cultured mouse entheseal tissue. As shown in Fig 1A, IL-17A up-regulated the level of *Col2A1* mRNA encoding type II collagen in the histoculture after 72 h. *Col2A1* is expressed in fibrocartilage entheses (Braun et al, 2000). The expression of *Acan* mRNA encoding aggrecan, a cartilage-specific proteoglycan core protein (Braun et al, 2000), was also augmented by IL-17A stimulation in a dose-dependent manner (Fig 1B). Moreover, IL-17A up-regulated the production of bone morphogenetic protein (BMP) 2, which is a crucial component of the BMP signaling cascade in endochondral bone formation (Lories & Luyten, 2009), in a dose-dependent manner (Fig 1C). Next, we used microarray analysis to identify the direct target gene of IL-17A in entheseal histocultures. Endochondral ossification–related genes, including BMP, Wnt, and FGF signaling genes (Salazar et al, 2016; Zhang et al, 2023), were up-regulated under IL-17A stimulation (Table 1). Among the candidate genes, IL-17A induced *FGF7*

expression (2.0-fold) in the culture (Table 1). We confirmed the significant increase in IL-17A–induced FGF7 production by both mRNA and protein quantification analyses in a dose-dependent manner (Fig 1D and E). Furthermore, as shown in Fig 1F and G, FGF7 up-regulated the expression of *Col2A1* and *Acan* in entheseal tissues for 3 h. The response time to elevated levels of the transcripts after FGF7 stimulation was shorter than that after IL-17A stimulation. The production of the BMP2 protein was induced by FGF7 stimulation in the culture for 24 h (Fig 1H). In addition, IL-17F, but not IL-17A, induced the up-regulation of *IL-17RC* expression after 24 h (Fig 1I). Neither IL-17A nor IL-17F up-regulated the expression of *IL-17RA* (data not shown).

### Inhibition of endochondral ossification by anti-FGFR2IIIb antibodies in entheseal histoculture

To confirm the inhibitory effect of the anti-FGFR2IIIb antibody on FGF7 signaling, we used 3T3-L1 cells as described previously (Zhang et al, 2010). In contrast to the RNA expression profiling in the study, flow cytometric analysis revealed the expression of FGFR2IIIb on the cell surface (Fig 2A). 3T3-L1 cells secreted IL-13 after FGF7 stimulation in a dose-dependent manner (data not shown), as well as IL-1β and TNFα stimulation (Kwon et al, 2014). To explore the blocking effect of the antibody, inhibition of IL-13 production in FGF7-stimulated 3T3-L1 cells was examined. As shown in Fig 2B, the anti-FGFR2IIIb antibody inhibited the release of IL-13 in a dose-dependent manner. The 50% inhibitory concentration ($IC_{50}$) of the antibody was 1.0 µg/ml. The anti-FGFR2IIIb antibodies significantly inhibited the expression of FGF7-stimulated transcripts of *Col2A1* and *Acan* in the entheseal histoculture at a maximum dose of 100 µg/ml (Fig 2C and D). Importantly, anti-FGFR2IIIb antibodies significantly inhibited the IL-17A–stimulated expression of *Col2A1* and *Acan* and the production of BMP2 in entheseal tissues at a dose of 100 µg/ml (Fig 2E–G). Surprisingly, the results of mRNA and protein quantification analyses showed a significant inhibition of FGF7 production by IL-17A stimulation at the maximum dose of the anti-FGFR2IIIb antibody (Fig 2H and I), indicating the presence of positive feedback regulation in FGF7 signaling, as described previously (Wearing & Sherratt, 2000). Collectively, these results indicate that IL-17A induced endochondral ossification via FGF7 production in entheseal tissues.

### Association between increased FGF7 and dermatitis and ankylosing arthropathy onsets in a PsA model

Aged male DBA/1J mice were used as a model of PsA (Lories et al, 2004). Male DBA/1J mice, when grouped after puberty, develop stress because of intermale aggressiveness and develop both psoriasis-like dermatitis and ankylosing enthesitis (Ebihara et al, 2015). This appears to resemble the "Koebner phenomenon" observed in human psoriasis, where environmental factors, such as physical traumatic injury or local stress, trigger disease onset (Stankler, 1969). Plasma FGF7 concentration was determined at 6, 18, 20, and 28 wk of age in male DBA/1J mice caged together; the points correspond to the young, subclinical, initial, and advanced stage of the observed dermatitis and arthropathy, respectively (Fig 3A). Furthermore, at 28 wk, the mice were divided into two

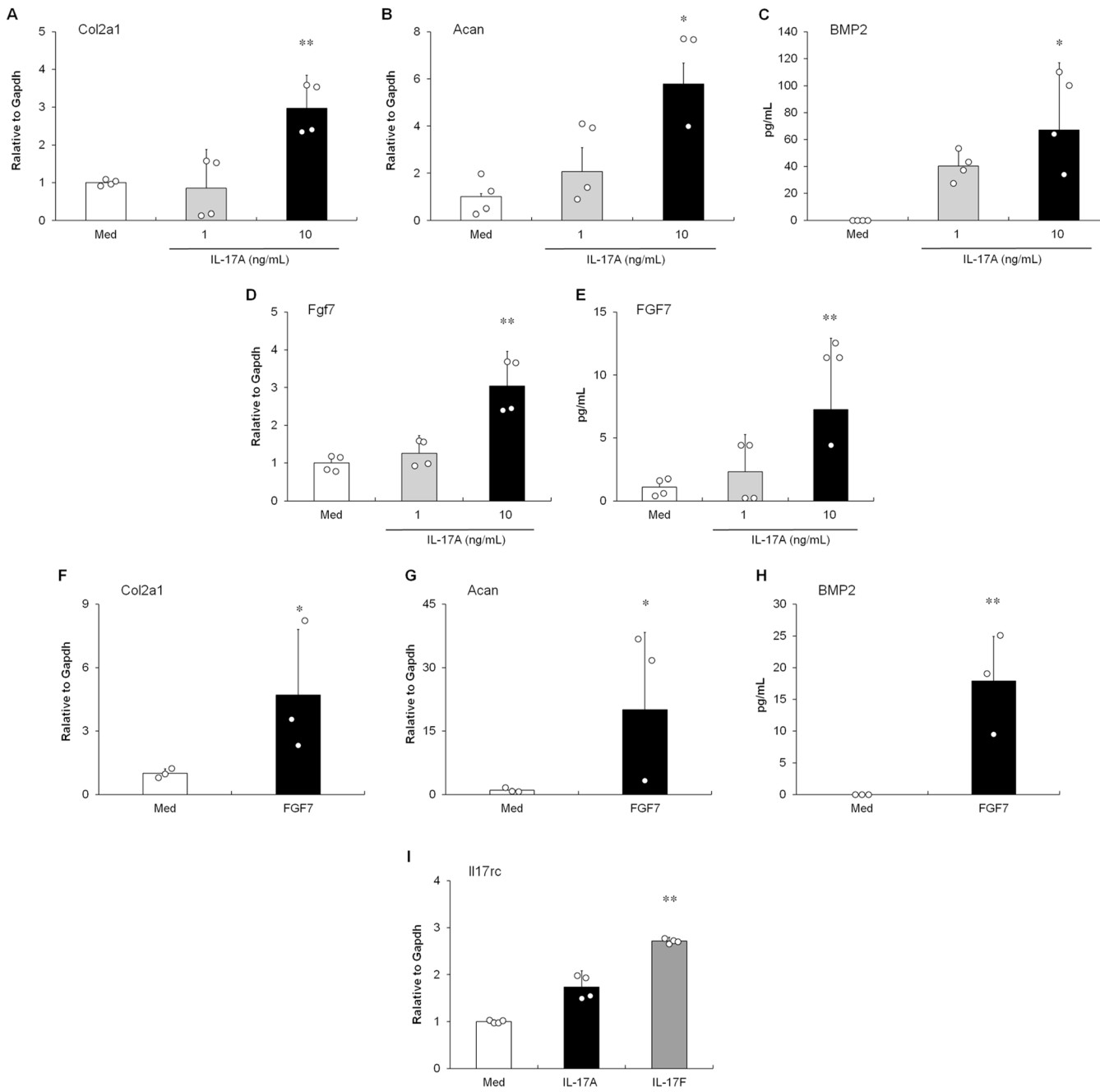

**Figure 1. IL-17A induced the up-regulation of endochondral ossification–related genes in the entheseal tissues.**
**(A, B)** Expressions of *Col2A1* (A) and *Acan* (B) mRNA in the entheseal tissue culture. mRNA levels were determined using quantitative RT–PCR 72 h after IL-17A stimulation. **(C)** Production of the BMP2 protein in the entheseal tissue culture supernatant. The protein level was determined using ELISA 72 h after IL-17A stimulation. **(D, E)** Levels of mRNA (D) and protein (E) of FGF7 in the histoculture 72 h after IL-17A stimulation. The protein level of FGF7 in the supernatant of the entheseal tissue was measured by ELISA. **(F, G)** Effect of FGF7 on the expressions of *Col2A1* (F) and *Acan* (G) mRNA in the tissue culture. The mRNA level was determined using quantitative RT–PCR 3 h after FGF7 stimulation (10 ng/ml). **(H)** Effect of FGF7 on the expressions of the BMP2 protein in the culture supernatant. The protein level was determined using ELISA 24 h after FGF7 stimulation (10 ng/ml). **(I)** Expression of *IL-17RC* mRNA in the histoculture. mRNA levels were determined using quantitative RT–PCR 24 h after IL-17A (10 ng/ml) or IL-17F (10 ng/ml) stimulation. The data presented are relative mRNA levels normalized to *GAPDH*. **(A, B, C, D, E, I)** Results are shown as the mean ± SD with individual data. *$P < 0.05$; **$P < 0.01$ by Dunnett's parametric test performed versus the non-stimulation group (Med). Each dot represents one experiment with pooled entheses from three to four mice. **(D, E)** *$P < 0.05$ by a *t* test or an Aspin–Welch *t* test performed versus the non-stimulation group (Med). Each dot represents one experiment with pooled entheses from three to four mice.

groups according to their disease status: affected and unaffected. As shown in Fig 3A, FGF7 in the plasma increased with the development of dermatitis and ankylosing enthesitis. Importantly, no increase was observed in the unaffected group at 28 wk. FGF10, another FGFR2IIIb ligand, was not detected in plasma (data not shown).

**Table 1. Endochondral ossification–related genes in the entheseal histoculture.**

| Gene symbol | Accession numbers | Fold change | Gene name |
|---|---|---|---|
| Grem1 | NM_011824 | 17.6 | Gremlin 1, DAN family BMP antagonist |
| Fgf10 | NM_008002 | 11.6 | Fibroblast growth factor 10 |
| Bmpr2 | ENSMUST00000186246 | 4.5 | Bone morphogenetic protein receptor, type II |
| Fgf1 | NM_010197 | 4.4 | Fibroblast growth factor 1 |
| Fgfr4 | NM_008011 | 4.0 | Fibroblast growth factor receptor 4 |
| Has1 | NM_008215 | 3.6 | Hyaluronan synthase 1 |
| Msx2 | NM_013601 | 3.1 | Msh homeobox 2 |
| Fgf6 | NM_010204 | 3.0 | Fibroblast growth factor 6 |
| Fgf18 | NM_008005 | 2.9 | Fibroblast growth factor 18 |
| Fgf23 | NM_022657 | 2.8 | Fibroblast growth factor 23 |
| Sox9 | NM_011448 | 2.0 | SRY (sex-determining region Y)-box 9 |
| Fgf7 | NM_008008 | 2.0 | Fibroblast growth factor 7 |
| Smad7 | NM_001042660 | 1.9 | SMAD family member 7 |
| Bmp4 | NM_007554 | 1.9 | Bone morphogenetic protein 4 |
| Wnt5b | NM_009525 | 1.9 | Wingless-type MMTV integration site family, member 5B |
| Smad1 | NM_008539 | 1.9 | SMAD family member 1 |
| Bmp2 | NM_007553 | 1.9 | Bone morphogenetic protein 2 |

Gene expression analysis of entheseal tissues. Organs were stimulated with or without IL-17A (10 ng/ml) for 72 h (n = 3 per sample). The criteria for the genes up-regulated by IL-17A stimulation were $P < 0.05$ and fold change > 1.8.

Next, we examined FGF7 expression in both the skin and joints. As shown in Fig 3B, the expression was significantly elevated in the joints, in the advanced stage compared with that in the young stage. In addition, FGF7 expression in the skin did not change in dermatitis, indicating constitutive FGF7 expression (Fig 3B). In contrast, the expression of FGF10 was not increased in either the skin or joints (data not shown).

## Local expressions of FGFR2 and IL-17A in the PsA model

To investigate the expression of FGFR2IIIb in the entheseal tissue of male DBA/1J mice, flow cytometry was used. The evaluation of FGFR2IIIb-positive cells in the entheses revealed that the FGFR2IIIb$^+$ fraction comprised CD45$^-$CD140a$^+$ cells from both the young and advanced stages of ankylosing enthesitis (Fig 3C–E). CD140a is a marker of mesenchymal stromal/stem cells (MSCs), which are capable of osteogenic and chondrogenic differentiation (Baustian et al, 2015). Regarding other markers of MSCs, including CD44 and Sca-1 (Baustian et al, 2015), FGFR2IIIb was expressed in any subpopulation of CD140a$^+$CD44$^+$, CD140a$^+$CD44$^-$, CD140a$^+$Sca-1$^+$, and CD140a$^+$Sca-1$^-$ (Fig S1). Thus, FGFR2IIIb is constitutively expressed in the entheses of the PsA model. In addition, the expression of FGFR1IIIb, which is another receptor for FGF7, was not detected in the joints (data not shown).

Next, to identify the common cellular sources of IL-17A secretion in both the skin and entheses at the advanced stage, we performed intracellular IL-17A staining in combination with various cell surface markers using flow cytometry. The results demonstrated that IL-17A originated from CD4$^+$ T cells (CD45$^+$TCR$\beta^+$CD4$^+$CD335$^-$), $\gamma\delta$ T cells

(CD45$^+$TCR$\beta^-\gamma\delta$TCR$^+$), and neutrophils (CD45$^+$TCR$\beta^-\gamma\delta$TCR$^-$CD11b$^+$Ly6G$^+$) in both the skin and entheses (Fig 3F–L). Moreover, the numbers of the infiltrating IL-17–positive cells in the entheses were increased compared with young mice (Fig 3M–O).

## Amelioration of the development and progression of ankylosing enthesitis but not dermatitis by the treatment of anti-FGFR2IIIb antibodies in the PsA model

To investigate the contribution of FGF7 signaling to the development of dermatitis and ankylosing enthesitis, FGF7 signaling was blocked by the systemic administration of an anti-FGFR2IIIb antibody. For intervention studies, anti-FGFR2IIIb antibodies were administered weekly to mice from 20 wk onward, when disease onset and FGF7 increased in the plasma. Treatment with the anti-FGFR2IIIb antibody resulted in significant suppression of both the severity and incidence of ankylosing enthesitis (Fig 4A and B). In addition, microscopic examination at the experimental endpoint revealed remarkable suppression of ankylosing enthesitis with endochondral ossification (Fig 4C–E). In contrast, the administration of the anti-FGFR2IIIb antibody had no effect on either the clinical or pathological changes in dermatitis (Fig 4F–I). Taken together, our results indicate that FGF7 signaling plays a critical role in both the development and progression of ankylosing enthesitis related to psoriatic skin lesions.

To monitor the plasma concentration of the anti-FGFR2IIIb antibody, blood was collected from these mice 1 wk after the final treatment with the antibody (at the experimental endpoint). The mean trough concentration in the anti-FGFR2IIIb antibody–treated

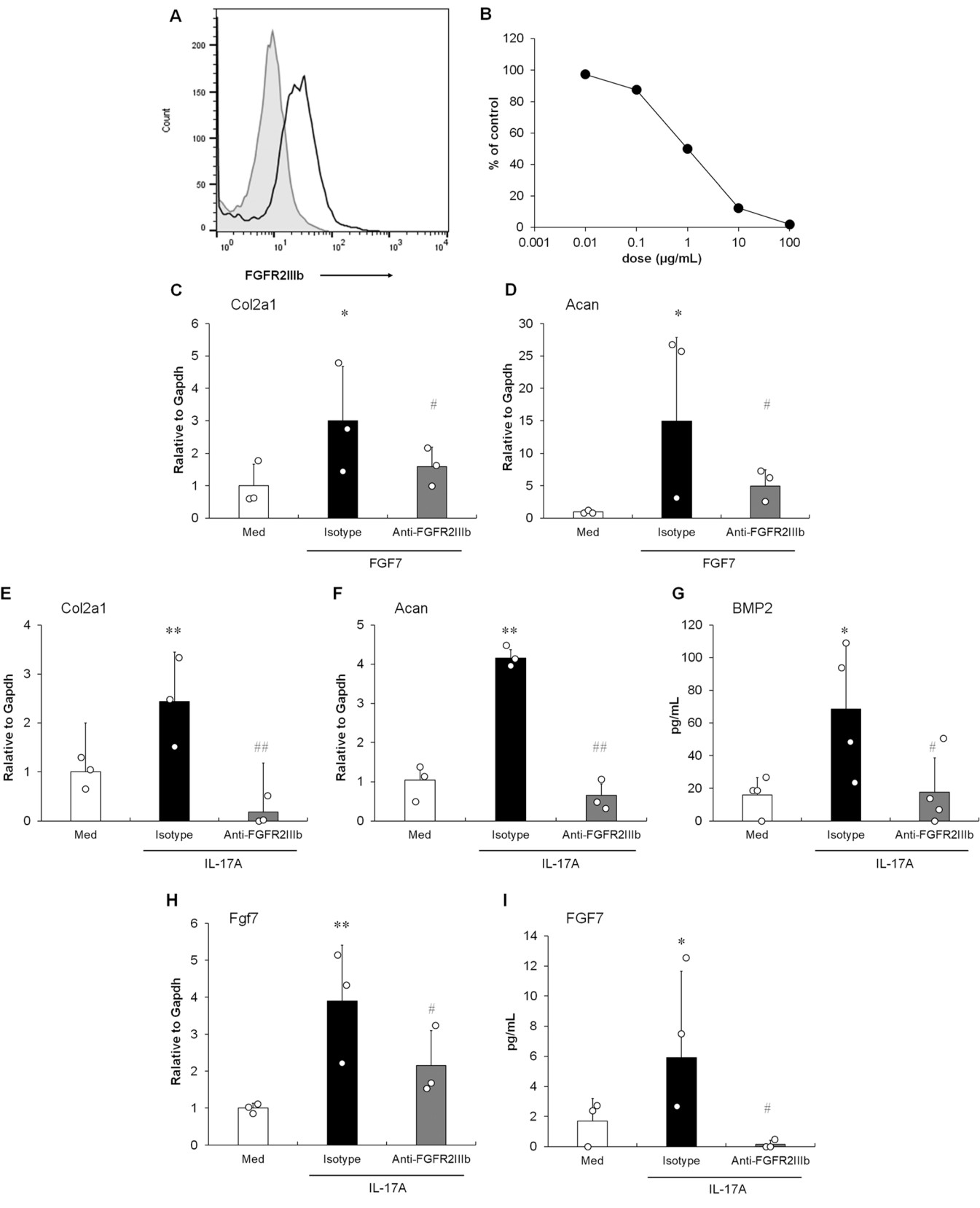

group was 62.8 μg/ml. FGF7-stimulated IL-13 production in 3T3-L1 cells was almost completely inhibited by this concentration, as calculated from the fitted sigmoid concentration–response curve (Fig 2B). These results indicate that the doses used in the PsA model were sufficient to show the therapeutic efficacy in ankylosing enthesitis.

### Effect of anti-FGFR2IIIb antibody treatment on pathologic responses in the PsA model

To explore the systemic effects of the anti-FGFR2IIIb antibody, plasma cytokines up-regulated with the development of dermatitis and ankylosing enthesitis were examined. Although both IL-17A and IL-17F, which belong to the IL-17 cytokine family, increased with the development of dermatitis and ankylosing enthesitis, anti-FGFR2IIIb treatment had no effect on cytokine production (Fig 5A and B). In contrast, antibody administration significantly decreased IL-6 levels in the plasma (Fig 5C). Taken together, these results indicate that IL-17A and IL-17F are upstream of FGF7 signaling and that IL-6 is downstream of the signaling pathway. As shown in Fig 5D, anti-FGFR2IIIb treatment also inhibited the systemic production of FGF7 in the PsA model and in entheseal histoculture (Fig 2H and I).

To examine the local effects of the anti-FGFR2IIIb antibody, cytokine-related transcript levels in both joints and skin were analyzed. Both *IL-17A* and *IL-17F* transcripts in the joint were up-regulated during ankylosing enthesitis development; however, anti-FGFR2IIIb treatment had no effect (Fig 5E and F). Moreover, the expression of *IL-6* and *FGF7* transcripts increased with ankylosing enthesitis, and anti-FGFR2IIIb antibodies inhibited the expression of these molecules (Fig 5G and H), as well as the results of plasma concentrations (Fig 5C and D). In addition, as shown in Fig 5I, the expression of *IL-17RC* also increased in the joints with pathological changes, suggesting that the up-regulation may be dependent on IL-17F stimulation, considering the results of the entheseal histoculture assay (Fig 1I). Taken together, these results indicate that cytokine expression in the joints was similar to systemic release. Meanwhile, as shown in Fig 5J and K, both *IL-17A* and *IL-17F* expression in the skin also increased with dermatitis, and the treatment with anti-FGFR2IIIb antibodies did not change. Unlike in whole blood (Fig 5C) and joints (Fig 5G), anti-FGFR2IIIb administration did not decrease the expression of the inflammatory cytokine *IL-6* in the skin (Fig 5L). In addition, the expression of psoriasis-related inflammatory response markers, including *Keratin 16, S100A8,* and *S100A9* (Ebihara et al, 2015), was not reduced by anti-FGFR2IIIb treatment (Fig 5M–O). It has been suggested that

FGFR2IIIb blockage does not influence the inflammatory functions of keratinocytes. Collectively, our results indicated that the development of dermatitis in this PsA model was independent of FGF7 signaling.

## Discussion

Several studies have indicated that IL-17A mediates bone destruction by up-regulating the receptor activator of nuclear factor-κB ligand (RANKL) and the induction of osteoclastogenesis (Sato et al, 2006; Adamopoulos et al, 2010). On the contrary, it has also been indicated that IL-17A plays a pivotal role in osteogenesis. Several studies have demonstrated that IL-17A induces osteoblast differentiation of human MSCs (Osta et al, 2014; Croes et al, 2016). However, an inhibitory effect of IL-17A on the differentiation of human MSCs into osteoblasts has also been reported (Wang et al, 2017). Therefore, the role of IL-17A in bone metabolism remains controversial.

The present study showed that IL-17A induced endochondral ossification in mouse organ cultures using entheseal tissues (Fig 1 and Table 1). The expression of *Col2A1* and *Acan* is up-regulated in the proliferation and differentiation of chondrocytes during the formation of the extracellular matrix as the initial phase of endochondral ossification (Clancy et al, 2003; Staines et al, 2013), leading to joint ankylosis (Lories & Haroon, 2017). The expression of *Col2A1* has been shown to increase significantly in the synovium of patients with PsA (Belasco et al, 2015). Although quantitative RT–PCR assays showed the augmentation of *Col2A1* and *Acan* expression in the histoculture under IL-17A stimulation (Fig 1A and B), up-regulation was not detected in the microarray, considering the differences caused by the limited sensitivity of gene arrays. In addition, BMP2 was released by IL-17A stimulation of the supernatant of the tissue culture (Fig 1C). BMP2, which is mainly produced by chondrocytes, plays a critical role in bone formation in an autocrine/paracrine manner (Clancy et al, 2003). BMP2 has been detected in synovial tissues of patients with SpA (Lories et al, 2003). Importantly, the inhibitory effect of the anti-FGFR2IIIb antibody in the histoculture showed that IL-17A–induced endochondral ossification was followed by the production of FGF7 in the enthesis (Fig 2E–G). The cellular source of FGF7 in enthesis may be MSCs or chondrocytes/perichondral cells, as described previously (Zhang et al, 2023). Taken together, our results indicate a novel function for IL-17A in bone formation, such as the induction of endochondral ossification.

---

**Figure 2.  Anti-FGFR2IIIb antibody showed an antagonistic effect on endochondral ossification in the entheseal histoculture.**
**(A)** Surface expression of FGFR2IIIb on 3T3-L1 cells was analyzed by flow cytometer. The gray line indicates the histogram obtained with an isotype control. **(B)** Effect of the anti-FGFR2IIIb antibody on IL-13 production in FGF7-stimulated 3T3-L1 cells. The concentrations of IL-13 in the supernatant were determined by a multiplex assay. Results are shown as the % of FGF7 stimulation. $IC_{50}$ was calculated in a sigmoid dose–response curve. **(C, D)** Effect of the anti-FGFR2IIIb antibody on the expressions of *Col2A1* (C) and *Acan* (D) mRNA by FGF7 stimulation. mRNA levels were determined using quantitative RT–PCR 3 h after FGF7 stimulation. **(E, F)** Effect of the anti-FGFR2IIIb antibody on the expressions of *Col2A1* (E) and *Acan* (F) mRNA by IL-17A stimulation. The mRNA levels were determined using quantitative RT–PCR 72 h after IL-17A stimulation. **(G)** Effect of the anti-FGFR2IIIb antibody on the BMP2 production by IL-17A stimulation in the histoculture. The protein level was determined using ELISA 72 h after IL-17A stimulation. **(H, I)** Levels of mRNA (H) and protein (I) of FGF7. The protein level of FGF7 in the supernatant of the entheseal histoculture was measured by ELISA. The data presented are relative mRNA levels normalized to *GAPDH*. **(C, D, E, F, G, H, I)** Results are shown as the mean ± SD with individual data. *P < 0.05; **P < 0.01 by a *t* test or an Aspin–Welch *t* test performed versus the non-stimulation group (Med). #P < 0.05; ##P < 0.01 by a *t* test or an Aspin–Welch *t* test performed versus the stimulation group (Isotype). Each dot represents one experiment with pooled entheses from three to four mice.

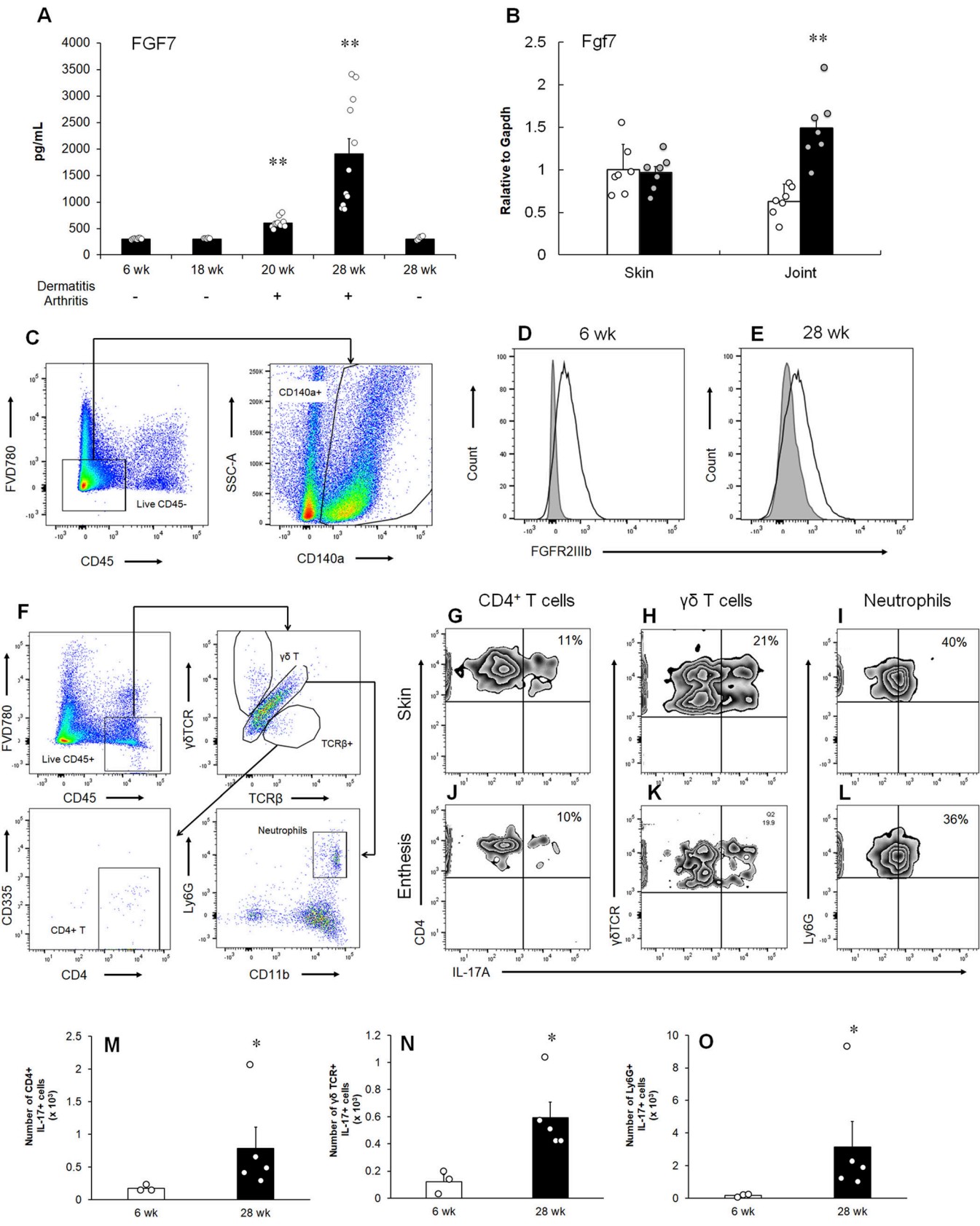

Among several models developed for both dermatitis and ankylosing enthesitis (Mandour et al, 2021), we selected male DBA/1J mice caged together, in which the symptoms were dependent on IL-17A (Ebihara et al, 2015) and human PsA (McInnes et al, 2015; Mease et al, 2017). The present study provides evidence of the role of FGF7 in the development of ankylosing enthesitis. The expression of FGF7 was up-regulated in the joints with the development of enthesitis (Fig 3B). In contrast, the expression of FGFR2IIIb was constitutive (Fig 3D and E). Ankylosing enthesitis was strongly inhibited by treatment with the anti-FGFR2IIIb antibody (Fig 4A–E). Dermatitis was not suppressed by the same treatment in the PsA model (Fig 4F–I). Unlike in the skin, production of both IL-6 and FGF7 in the joints was inhibited by anti-FGFR2IIIb antibody administration (Fig 5G and H), as well as in whole blood (Fig 5C and D), suggesting that the antibody showed an antagonistic effect at the joint and that the cytokines leaked from the joint into whole blood. FGF7 was not up-regulated in the whole blood at the subclinical stage (Fig 3A), when IL-17A production was increased (data not shown), as in our previous report (Ebihara et al, 2015). Thus, IL-17A release in whole blood is followed by FGF7 production and disease onset. The infiltration of IL-17A–positive CD4$^+$ T cells, $\gamma\delta$ T cells, and neutrophils was observed in the skin at the experimental endpoint (Fig 3G–I) and in human psoriasis (Kryczek et al, 2008; Cai et al, 2011; Dyring-Andersen et al, 2017), suggesting a critical role of the damaged skin–derived immune cells in systemic IL-17A production. Furthermore, because the infiltration of these cells was also observed in the joints and in human SpA and PsA (Leipe et al, 2010; Appel et al, 2011; Guggino et al, 2016), the up-regulation of IL-17A transcripts (Fig 5E) is thought to be detected in infiltrating CD4$^+$ T cells, $\gamma\delta$ T cells, and/or neutrophils (Fig 3J–L). The mechanisms of migration of skin-derived IL-17A–positive cells need to be further investigated. In addition, it will also be important to determine whether IL-17A produced from tissue sources other than the skin contributes to the pathological connection between the skin and joints. IL-17A–producing $\gamma\delta$ T cells (CD3$^+$CD4$^-$CD8$^-$$\gamma\delta$TCR$^+$) have been reported to be enthesis-resident in IL-23 minicircle–induced SpA models (Reinhardt et al, 2016). Nevertheless, our present findings suggest that IL-17A is produced in cutaneous inflammation caused by the aggressive behavior of male DBA/1J mice caged together, resulting in stimulation of the enthesis, subsequently resulting in elevated local levels of FGF7, and the induction of both further inflammation and endochondral ossification via FGF7 signaling. Physical trauma, termed the deep Koebner phenomenon, is a potential trigger for PsA (McGonagle et al, 2011). Thus, the expression of FGF7 at enthesis might also be up-regulated not only by skin inflammation, but also by trauma in human PsA. In a previous study, we defined the FGFR2 coding locus as an ankylosis

susceptibility locus in (MRL/rpl × C3H/lpr) F1 mice, which develop spontaneous ankylosing arthropathy, as well as DBA/1J mice (Mori et al, 2006). Other findings also suggested that IL-17F, which is up-regulated in both the skin and synovium of patients with PsA (Sánchez-Rodríguez & Puig, 2023), has been produced by both dermatitis and ankylosing enthesitis and has augmented IL-17A and/or IL-17F signaling through the increase of IL-17RC expression in the enthesis (Figs 1I and 5I).

Although recent studies on the pathology of PsA have revealed its association with psoriasis, several questions regarding the mechanisms underlying the direct link between psoriatic skin and joint inflammation remain to be answered (Belasco et al, 2015; Winthrop et al, 2019). One unresolved question is whether skin inflammation triggers the development of musculoskeletal features. Another is the role of IL-17A in bone erosion and formation. The present findings suggest a pathological connection between the skin and joints in patients with PsA. FGF7, which has been produced in the enthesis by systemic circulating damaged skin–derived IL-17A stimulation, plays a critical role in endochondral ossification and ankylosis. Therefore, it has been suggested that the anti-IL-17A antibody is effective against PsA, causing osteogenesis of the joint (McInnes et al, 2015; Mease et al, 2017), but not rheumatoid arthritis caused by bone and cartilage destruction (Genovese et al, 2013).

Our findings further indicated that anti-FGFR2IIIb is a promising therapy for joint lesions in PsA. The FGFR2IIIb-blocking antibody bemarituzumab (formerly FPA144; Five Prime Therapeutics) is currently undergoing phase II clinical trials against gastric cancer (FIGHT study) (NCT03694522) and is expected to be a therapeutic agent for PsA. In summary, we propose FGF7 as an essential mediator for the development of ankylosing arthropathy related to psoriatic skin lesions in the pathological connection in a mouse PsA model and provide novel insight into the therapeutic approach of human PsA.

# Materials and Methods

### Mice

Male C57BL/6J (B6J) and DBA/1J mice were purchased from the Jackson Laboratory Japan and maintained under specific pathogen-free conditions at the Animal Research Institute of Tohoku University Graduate School of Medicine, Sendai, Japan. All experiments performed in this study conformed to the ethical guidelines of the Tohoku University for animal experimentation.

---

**Figure 3. Association between FGF7 and FGFR2IIIb expressions with the onset of dermatitis and ankylosing enthesitis.**
**(A)** Plasma concentrations of FGF7 were determined by ELISA at the indicated ages (wk). **(B)** Expression of *FGF7* mRNA in both the skin and joints at the young (white column) and advanced stages (black column). The mRNA levels were determined using quantitative RT–PCR. n = 7 animals per group. **(C)** Gating strategy for analysis of FGFR2IIIb-expressing cells. **(D, E)** FGFR2IIIb expression of entheses cells of the young (D) and advanced stages (E). Gray lines indicate the histogram obtained with isotype controls. Data are representative of the stained entheseal cells of five mice. **(F)** Gating strategy for analysis of IL-17A–producing cells. **(G, H, I, J, K, L)** Intracellular IL-17A staining of skin (G, H, I) and entheseal (J, K, L) cells. **(G, H, I, J, K, L)** CD4$^+$ T cells (G, J), $\gamma\delta$ T cells (H, K), and neutrophils (I, L). **(M, N, O)** Number of infiltrating cells into enthesis at the young (white column) and advanced stages (black column). **(M, N, O)** CD4$^+$ T cells (M), $\gamma\delta$ T cells (N), and neutrophils (O). **(A, B, M, N, O)** Results are shown as the mean ± SEM with individual data. **(A)** **P < 0.01 by Dunnett's test performed versus the young stage group (6 wk). n = 5–11 animals per group. **(B)** **P < 0.01 by an Aspin–Welch test performed versus the young stage group. n = 7 animals per group. **(M, N, O)** *P < 0.05 by an Aspin–Welch test performed versus the young stage group. n = 3–5 animals per group.

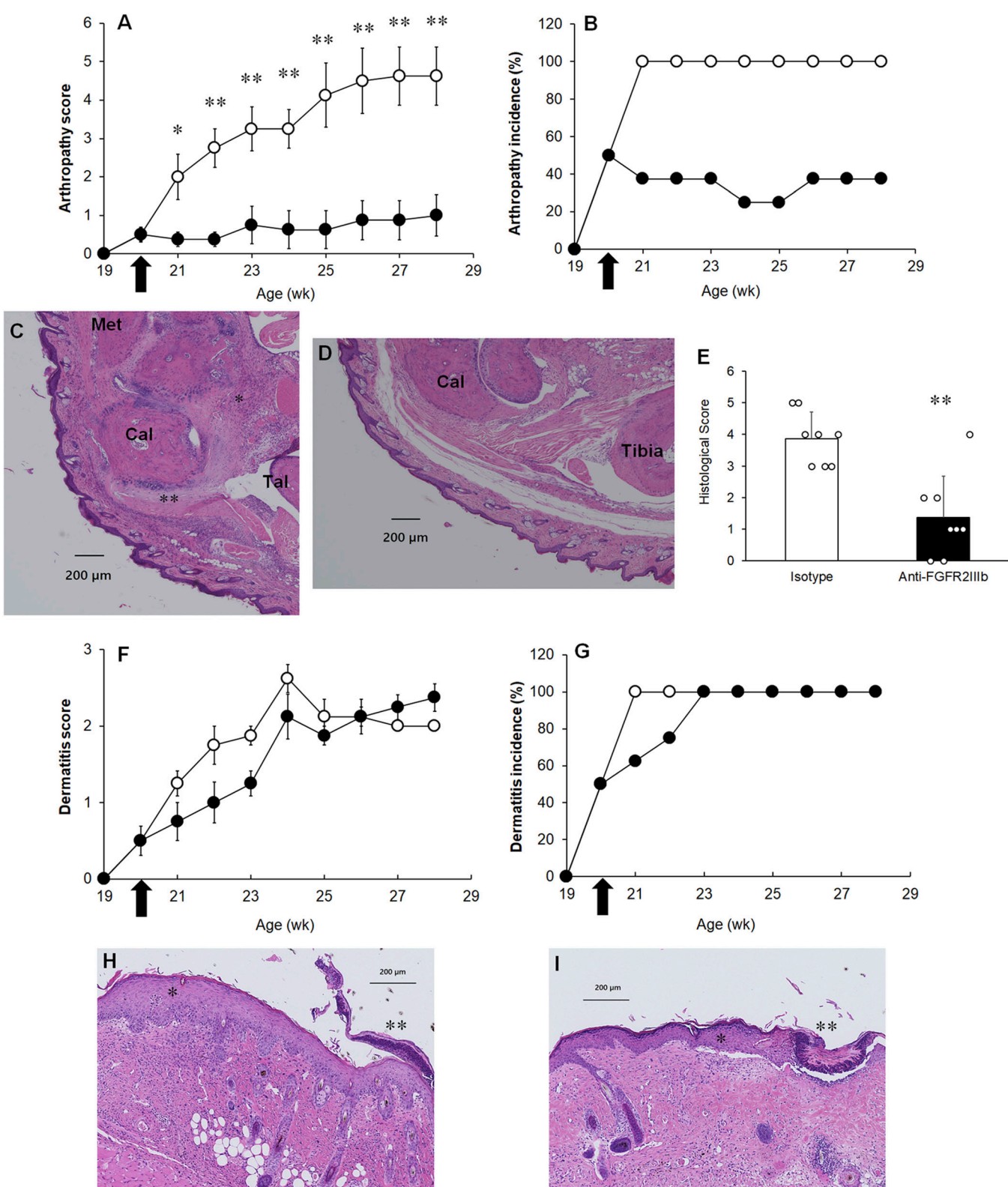

**Figure 4. Effects of administration of the anti-FGFR2IIIb antibody on the progression of dermatitis and ankylosing enthesitis.**
**(A, B)** Clinical score (A) and incidence (B) of spontaneous arthropathy of two groups: isotype control antibody–treated (open circles) and anti-FGFR2IIIb antibody–treated (closed circles) groups. *$P < 0.05$; **$P < 0.01$ by the Mann–Whitney $U$ test performed versus the isotype control antibody–treated group. **(B)** Incidence of arthropathy was 100% (8/8) in isotype control antibody–treated (open circles) and 37.5% (3/8) in anti-FGFR2IIIb antibody–treated (closed circles) groups at the experimental endpoint (28 wk). The $P$-value was calculated as 0.026 by Fisher's exact test; the difference was considered significant. **(C, D, E)** Representative

## Cell culture

The murine 3T3-L1 preadipocyte cell line was obtained from the Japan Health Sciences Foundation and maintained in a growth medium consisting of DMEM and 10% FBS in a humidified atmosphere containing 5% $CO_2$ at 37°C.

## Cellular treatments

Allophycocyanin (APC)-conjugated anti-rat IgG (R&D Systems) was used to stain rat anti-FGFR2IIIb antibody (clone 133730; R&D Systems). Rat IgG2a (clone 2A3; Bio X Cell) was used as the isotype control. Cell surface staining of 3T3-L1 cells was performed according to standard techniques. Flow cytometry was performed using a FACSCalibur flow cytometer (BD Biosciences) and analyzed using FlowJo software (BD Biosciences).

3T3-L1 cells were seeded in 96-well plates at a density of $1 \times 10^4$ cells/well. Cells were starved for 24 h and then treated with 30 ng/ml FGF7 (R&D Systems). To examine the antagonistic effect of the anti-FGFR2IIIb antibody on FGF7-induced IL-13 production, the antibody was added 1 h before the addition of FGF7 (R&D Systems). Cells were cultured for 24 h, and the supernatant was collected and stored at −80°C until assayed. IL-13 was detected using MILLIPLEX MAP Kit according to the manufacturer's instructions (Merck Millipore).

## Entheseal histoculture

Entheseal tissues were cultured as previously described (Sherlock et al, 2012) with some modifications. In brief, to obtain entheseal tissues, male B6J mice (25 wk) were euthanized under anesthesia, their hind paw skins were removed, and the Achilles tendon and plantar fascia were traced to the point of insertion (four tissues per mouse). The tissue (3 mm) was obtained by cutting the tendon insertion at the surface of the bone under sterile conditions. Tissue fragments were plated in a 96-well plate in RPMI 1640 supplemented with 10% FBS, penicillin (100 units/ml), and streptomycin (100 U/ml). The organs were cultured with IL-17A (10 ng/ml; R&D Systems), IL-17F (10 ng/ml; R&D Systems), or FGF7 (10 ng/ml). In experiments examining the inhibitory effect of the anti-FGFR2IIIb antibody on the IL-17A–induced expression of *Col2A1* or *Acan*, production of BMP2 and FGF7, FGF7-induced expression of *Col2A1* or *Acan*, and production of FGF7, the antibody was added 1 h before the addition of IL-17A or FGF7. Rat IgG2a was used as the isotype control. Entheseal tissues were cultured for 24 h (FGF7 simulation) or 72 h (IL-17A stimulation), and the supernatants were collected and stored at −80°C until they were assayed. The amounts of FGF7

(Wuhan Huamei Biotech) and BMP2 (R&D Systems) were determined using an ELISA kit. For RNA analysis, entheseal tissues were cultured for 3 h (FGF7 simulation) or 72 h (IL-17A or IL-17F stimulation).

## RNA isolation

Total RNA was isolated from the entheseal tissue culture samples using RNeasy Plus Universal Mini Kit (QIAGEN) according to the manufacturer's instructions. RNA samples were quantified using an ND-1000 spectrophotometer (NanoDrop Technologies), and the quality was checked using Agilent 2200 Tape Station.

## Gene expression microarray and data analysis

The cRNA was amplified and labeled using Low Input Quick Amp Labeling Kit (Agilent Technologies) and hybridized using SurePrint G3 Mouse Gene Expression Microarray 8 × 60 K v2 (Agilent Technologies). All hybridized microarray slides were scanned using an Agilent scanner. Relative hybridization intensities and background hybridization values were calculated using Agilent Feature Extraction Software (9.5.1.1). The raw signal intensities and flags for each probe were calculated from hybridization intensities and spot information, according to the procedures recommended by Agilent Technologies, using the Flag criteria in GeneSpring software. The raw signal intensities of triplicate samples were normalized by quantile algorithm using the "preprocessCore" library package on Bioconductor software. Probes that call the "P" flag in at least one sample were selected, excluding lincRNA probes. We then applied the Linear Models for Microarray Analysis (limma) package of Bioconductor software for expression analysis between the controls (no stimulation) and experimental samples (IL-17A stimulation) and obtained *P*-values and fold changes for each gene. The criteria for the regulated genes were as follows: $P < 0.05$, ratio > 1.8 (up-regulated genes), and $P < 0.05$, ratio < 0.56 (down-regulated genes). Microarray data analysis was supported by Cell Innovator. Our data were deposited in the Gene Expression Omnibus database (accession number GSE234831).

## Real-time PCR

Quantitative reverse transcription (RT)–PCR was performed using the RNA-to-CT 1-step kit (Thermo Fisher Scientific) and QuantStudio 7 Flex Real-Time PCR System (Thermo Fisher Scientific). The primers (*Col2A1*: Mm01309565_m1, *Col4A1*: Mm01210125_m1, *FGF7*: Mm00433291_m1, *Acan*: Mm00545794_m1, *FGF10*: Mm00433275_m1, *IL-17RA*: Mm00434214_m1, *IL-17RC*: Mm00506606_m1, *IL-17A*: Mm00439618_m,

---

microphotograms of ankle joint obtained from isotype control antibody–treated (C) or anti-FGFR2IIIb antibody–treated (D) mice (H&E stain). Fibroblast-like cell differentiation with chondrocytic differentiation (asterisk) and ossification (double asterisk) is shown. Met, metatarsal bone; Cal, calcaneus; Tal, Talus bone. Scale bar, 200 μm. **(E)** Histological score of enthesitis between isotype control antibody–treated (white column) and anti-FGFR2IIIb antibody–treated (black column) mice at the experimental endpoint. Results are shown as the mean ± SEM with individual data. **P < 0.01 by the Mann–Whitney *U* test performed versus the isotype control antibody–treated group. **(F, G)** Clinical score (F) and incidence (G) of spontaneous dermatitis of two groups: isotype control antibody–treated (open circles) and anti-FGFR2IIIb antibody–treated (closed circles) groups. **(H, I)** Representative microphotograms of tail base skin obtained isotype control antibody–treated (H) or anti-FGFR2IIIb antibody–treated (I) mice (H&E stain). In (H, I), acanthosis with hyperkeratosis (asterisk) and crust formation (double asterisk) is shown. Scale bar, 200 μm. **(A, F)** Results are shown as the mean ± SEM. n = 8 animals per group. **(A, B, F, G)** Antibodies were administrated weekly at a dose of 200 μg/mouse beginning from 20 wk (arrows) to 1 wk before the experimental endpoint (27 wk).

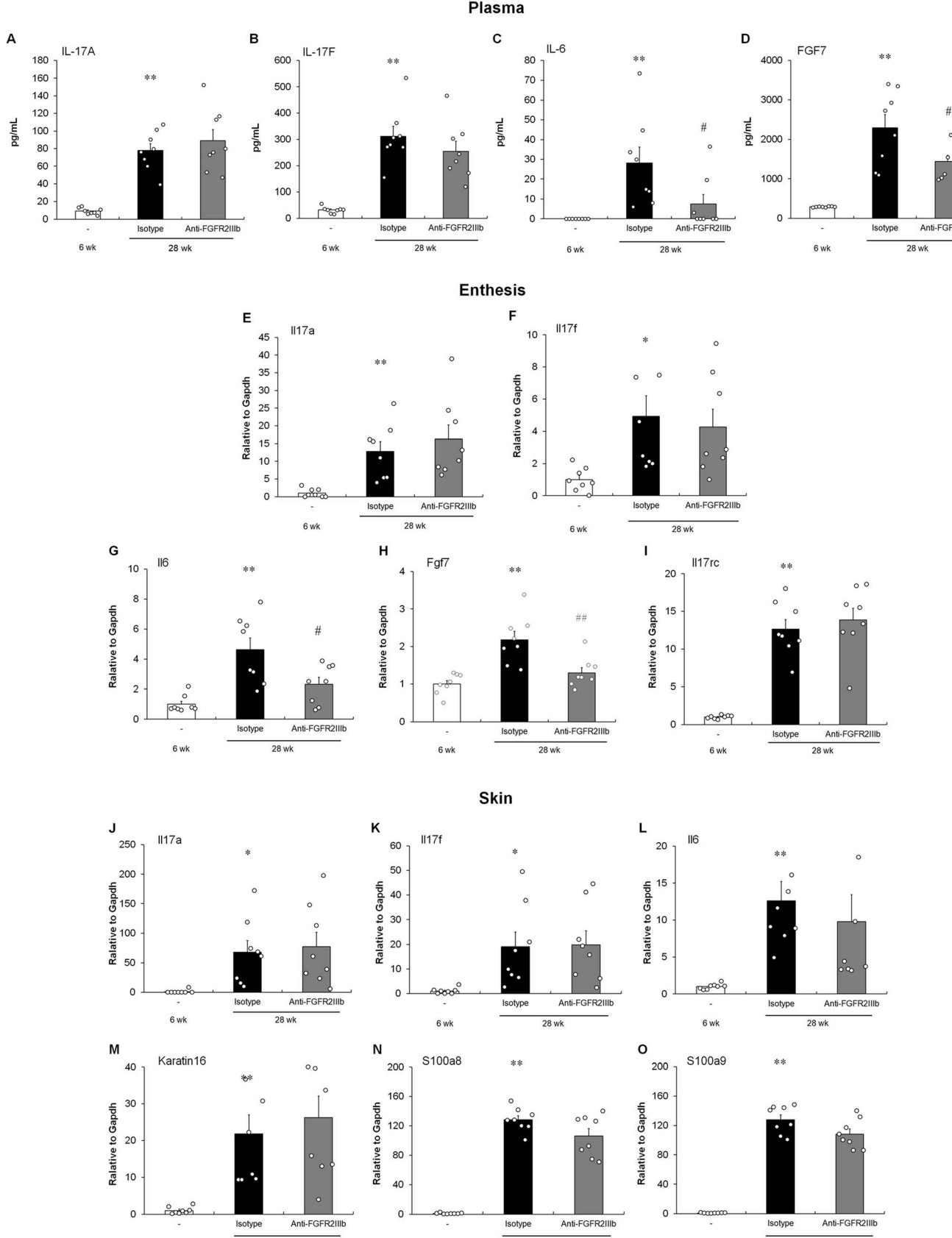

**Plasma**

A  IL-17A

B  IL-17F

C  IL-6

D  FGF7

**Enthesis**

E  Il17a

F  Il17f

G  Il6

H  Fgf7

I  Il17rc

**Skin**

J  Il17a

K  Il17f

L  Il6

M  Karatin16

N  S100a8

O  S100a9

*IL-17F*: Mm00521423_m1, *IL-6*: Mm00446190_m1, *Keratin 16*: Mm01306670_g1, *S100A8*: Mm00496696_m1, *S100A9*: Mm00656925_m1, *FGFR1IIIb*: Mm01215492_m1, and *GAPDH*: Mm99999915_g1) were from Thermo Fisher Scientific. The threshold cycle (Ct) generated by the quantitative RT–PCR system was used, and the mRNA expression levels were normalized to the level of *GAPDH* expression according to the $2^{-\Delta Ct}$ method.

## Preparation of cells in both entheses and skins, and stimulation

Entheseal tissues and tail base skins were digested for 1 h in RPMI containing 10% FCS, 0.33 mg/ml Liberase TL (Sigma-Aldrich), and 0.5 mg/ml DNase (Sigma-Aldrich). Digestion was stopped by adding 0.01 M EDTA. The cells were stimulated with PMA (Sigma-Aldrich) and ionomycin (Sigma-Aldrich) for 4 h in the presence of brefeldin A (Thermo Fisher Scientific).

## Flow cytometric analysis

Biotinylation of the anti-FGFR2IIIb antibody was performed using the Biotin-XX Microscale Protein Labeling kit (Thermo Fisher Scientific) according to the manufacturer's instructions. The antibodies against CD45, CD45RA (B220), CD8b, and CD44 were purchased from BD Biosciences. Antibodies against CD11b, CD4, Ly6G, TCRγδ, TCRβ, CD11c, CD335 (NKp46), Sca-1, and CD140a were purchased from BioLegend. Antibodies against IL-17A were purchased from Thermo Fisher Scientific. Streptavidin (BioLegend) was used to stain the biotin-labeled antibody. Cell surface staining was performed according to standard techniques after treatment with an anti-CD16/32 antibody (BD Biosciences) to block FcγR binding. Dead cells were excluded using Fixable Viability Dye eFluor 780 (FVD780; Thermo Fisher Scientific). For intracellular staining, cells were first stained with different cell surface antibodies, fixed, permeabilized, and intracellularly stained for IL-17A. Gating strategies were set with reference to the isotype or fluorescence minus one control. Flow cytometry was performed using a BD LSRFortessa X-20 flow cytometer (BD Biosciences) and analyzed using FlowJo software, version 10.8.0 (BD Biosciences).

## Determination of the effect of the anti-FGFR2IIIb antibody on the spontaneous PsA model

Male DBA/1J mice from different litters were mixed and caged in groups of eight mice at 20 wk after the onset of both arthropathy and dermatitis, based on plasma FGF7 levels. Mice were treated with 200 µg of the anti-FGFR2IIIb antibody or rat IgG2a by intraperitoneal injection once a week for 8 wk, until 1 wk before the experimental endpoint, as described previously (Ebihara et al, 2015). After euthanizing the mice under anesthesia at the experimental endpoint (28 wk), hind paws and skins at the tail base were removed and

frozen for RNA extraction or fixed for histological analysis. Plasma samples were stored at –80°C until they were assayed. ELISA kits were used to measure IL-17A (R&D Systems), IL-17F (R&D Systems), IL-6 (R&D Systems), FGF10 (Wuhan Huamei Biotech), and FGF7 levels in the plasma. RNA isolation from the paws or skin and quantitative RT–PCR were performed as already described.

## Pathological evaluation of both arthropathy and dermatitis

Mice were scored weekly for clinical signs of arthropathy as follows: 0 (no symptoms), 1 (redness and swelling in one toe), 2 (redness and swelling in more than one toe), 3 (toe stiffness), and 4 (deformity or ankle involvement), as previously described (Lories et al, 2005). Both hind paws were evaluated, which resulted in a maximum score of eight. The severity of dermatitis at the tail base was scored weekly as hair loss and crust formation according to the following grading convention: 0 (none), 1 (slight), 2 (moderate), 3 (severe), and 4 (very severe). For microscopic examination, after killing the mice under anesthesia, the hind paws and skin at the tail base were removed, fixed in 10% formalin in PBS (pH 7.2), and embedded either directly in paraffin wax or after decalcification (bone) in 5% formic acid containing 5% formalin. Tissue sections were stained with hematoxylin and eosin. The preparation of tissue specimens for pathological evaluation was supported by the Pathology Platform of the Tohoku University. The hind paw ankles were scored according to the histological criteria as follows: 0 (normal ankle), 1 (acute inflammation), 2 (slight heterotopic proliferation of cartilage or fibrocartilage tissue), 3 (severe heterotopic proliferation of cartilage or fibrocartilage tissue), and 4 (bone formation and ankylosis). The scoring systems of the clinical features of the disease and histology were determined by an investigator blinded to the treatment.

## Measurement of anti-FGFR2IIIb antibody levels in the plasma

The anti-FGFR2IIIb antibody in the plasma was detected using ELISA. Briefly, diluted plasma samples and standards were added to wells pre-coated with recombinant mouse FGFR2 (Fc Tag) (Sino Biological) and incubated at room temperature for 2 h. After washing with PBS/Tween-20, diluted HRP-conjugated anti-rat IgG (Thermo Fisher Scientific) was added to each well and incubated for 2 h. After washing, the TMB substrate was added to each well and incubated in the dark. The stop solution was added to each well, and the absorbance was measured at 450 nm.

## Statistical analyses

The significance of the average differences between the two groups was evaluated using the F test followed by a *t* test or an Aspin–Welch *t* test. The significance of average differences among multiple

**Figure 5. Effect of administration of the anti-FGFR2IIIb antibody on inflammation responses in the PsA model.**
**(A, B, C, D)** ELISA kits were used to measure IL-17A (A), IL-17F (B), IL-6 (C), and FGF7 (D) in the plasma at the experimental endpoint. **(E, F, G, I)** Quantitative RT–PCR was used to measure *IL-17A* (E), *IL-17F* (F), *IL-6* (G), *FGF7* (H), and *IL-17RC* (I) in the ankle joint at the experimental endpoint. **(J, K, L, M, N, O)** Quantitative RT–PCR was used to measure *IL-17A* (J), *IL-17F* (K), *IL-6* (L), *Keratin 16* (M), *S100A8* (N), and *S100A9* (O) in the skin of the tail base at the experimental endpoint. Results are shown as the mean ± SEM with individual data. *$P < 0.05$; **$P < 0.01$ by a *t* test or an Aspin–Welch *t* test performed versus the young group (6 wk). #$P < 0.05$; ##$P < 0.01$ by a *t* test or an Aspin–Welch *t* test performed versus the isotype control antibody (Isotype). n = 8 animals per group.

groups was evaluated using Dunnett's test. Differences in disease severity were evaluated using the Mann–Whitney *U* test. Fisher's exact test was used for incidence. *P* < 0.05 was considered significant. We performed statistical analysis using StatLight version 6.31 or Pharmaco Basic version 17.

## Data Availability

The microarray data were deposited in Gene Expression Omnibus [GEO: GSE234831]. All data that support the findings of this study are available from the corresponding author on reasonable request. On request, the corresponding author will provide the data that support the findings of this study.

## Supplementary Information

## Acknowledgements

We appreciate Dr. Masato Nose (Tohoku University Graduate School of Medicine) for his critical comments on the article and Mr. Yutaka Kurabe (Mito Medical Center) for assistance with tissue processing. This study was supported by grants-in-aid for scientific research from the Ministry of Education, Science, Sports and Culture of Japan to M Ono (No 23659198).

### Author Contributions

S Ebihara: conceptualization, data curation, formal analysis, validation, investigation, methodology, and writing—original draft, review, and editing.
Y Owada: methodology.
M Ono: conceptualization, resources, data curation, formal analysis, supervision, funding acquisition, validation, project administration, and writing—review and editing.

### Conflict of Interest Statement

The authors declare that they have no conflict of interest.

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
