## [Reviewer comments · Life Science Alliance]

Life Science Alliance

FGF7 as an essential mediator for the onset of ankylosing enthesitis related to psoriatic dermatitis

Shin Ebihara, Yuji Owada and Masao Ono
DOI: <https://doi.org/10.26508/lsa.202403073>

Corresponding author(s): Dr. Shin Ebihara (Tohoku University Graduate School of Medicine)

Review Timeline:

Submission Date:	2024-10-02
Editorial Decision:	2024-12-13
Revision Received:	2025-01-28
Editorial Decision:	2025-01-29
Revision Received:	2025-01-31
Accepted:	2025-01-31

Transaction Report:

December 13, 2024

Re: Life Science Alliance manuscript #LSA-2024-03073

Dr. Shin Ebihara
Tohoku University Graduate School of Medicine
Department of Pathology
2-1 Seiryō, Aoba
Sendai, Miyagi 980-8575
Japan

Dear Dr. Ebihara,

Thank you for submitting your manuscript entitled "FGF7 as an essential mediator for the onset of ankylosing enthesitis related to psoriatic dermatitis" to Life Science Alliance. The manuscript was assessed by expert reviewers, whose comments are appended to this letter. We invite you to submit a revised manuscript addressing the Reviewer comments.

Thank you for this interesting contribution to Life Science Alliance. We are looking forward to receiving your revised manuscript.

Sincerely,

B. MANUSCRIPT ORGANIZATION AND FORMATTING:

Reviewer #1 (Comments to the Authors (Required)):

This is an interesting and straightforward study showing a link between IL-17A and FGF7 expression in the joint. However, the 'emphasis / suggestion' of the authors that "IL-17A in PsA dermatitis induces the elevation of FGF7 levels in joint enthesitis" is not the whole story as IL-17A can also be produced directly in the joint. The authors do not prove that IL-17A is coming from the skin and that the skin is the (only) source of IL-17A to upregulate FGF7 in the joint causing ankylosing enthesitis. Therefore, this 'message/suggestion' in the manuscript should be rewritten.

Reviewer #2 (Comments to the Authors (Required)):

In this manuscript, Ebihara et al demonstrate a role for FGF7 in joint inflammation in a mouse model of psoriatic arthritis (DBA/1J). First, they demonstrate that IL-17A, which is known to play an important role in psoriasis, induces FGF7 expression at both the RNA and protein level in joint tissue cultures from mice. They next demonstrate that FGF7 induces Col2a1, Acan and BMP expression, which are associated with ossification. Importantly, they demonstrate both in vitro and in vivo antibody blockade of the FGF7 receptor FGFR2IIIb reduces expression of these genes and reduces arthropathy incidence, respectively. FGFR2IIIb blockade had no effect on IL17 family member expression, but did reduce IL6 and Fgf7 expression in the joint, indicating that the FGF7 effects are all downstream of IL17. There was no effect of FGFR2IIIb blockade on skin disease. This is an important study that begins to tease apart the different pathogeneses of skin versus joint involvement in psoriatic arthritis. I have a few comments that should be addressed:

Major:

- in all figure legends, please indicate the number of experimental replicates, as it is unclear how many times each experiment was performed (3-4 tissues, but from how many separate experiments/batches/mice?). A minimum of two to three repeat experiments from separate biological donors is customary in the field.
- Suggest making a summary figure demonstrating the pathway and joint-specific findings

Minor:

- lines 47-48 p5: suggest rewording to "The prevalence of PsA in the psoriasis population varies widely, most recently estimated to range from 5 to 40%"
- line 156 p 11, maybe consider mentioning koebnerization here?
- line 287 p 19 missing the i for in
- line 302 missing space between synovium of

Reviewer #3 (Comments to the Authors (Required)):

I would like to commend the authors, Ebihara et al., for their significant effort in elucidating the link between IL-17A and FGF7 in Psoriatic Arthritis (PsA). They have demonstrated distinct mechanisms occurring in psoriatic skin and ankylosing enthesitis and highlighted the potential therapeutic role of inhibiting the FGF receptor to ameliorate ankylosing enthesitis in PsA.

Psoriatic skin exhibits an increase in IL-17A levels, which further promotes the infiltration of CD4+ T cells, $\gamma\delta$ T cells, and neutrophils. Elevated IL-17A levels are also observed in PsA conditions.

The authors have convincingly shown increased Col2A and Acan expression using a novel histoculture technique. Additionally, they demonstrated that IL-17A stimulation leads to the release of BMP-2, which facilitates bone formation, as evidenced by the in vitro histoculture method. To explore the mechanisms linking IL-17A to PsA, the authors conducted a microarray analysis and identified the role of FGF7 (KGF). By targeting KGF using an antagonist of its receptor, FGFR2IIIb, they were able to inhibit ankylosing enthesitis in a PsA mouse model, though this effect was not observed in dermatitis. Furthermore, blocking the KGF receptor reduced the levels of IL-6 and KGF in joints, which were responsible for systemic elevation in whole blood.

I have some minor comments and hope the authors can address these through additional experiments or discussion:

1. FGF7/KGF Binding Specificity

FGF7/KGF is known to bind not only FGFR2IIIb but also FGFR1IIIb. It would be important to perform the same experiments using an FGFR1 antibody to confirm whether FGF7 and IL-6 levels decrease as hypothesized. Testing with SU5402, an FGFR1 inhibitor, may also provide further insights.

2. Alternative Inhibitors for FGFR2IIIb

Instead of receptor-blocking antibodies like FGFR2IIIb, it would be valuable to evaluate the efficacy of small molecules, such as TAS-120 (Futibatinib), or ligand traps (if available) for FGFR2IIIb. If similar results are obtained, it would support the utility of these small molecules in therapeutic applications.

28 Jan 2025

Re: LSA-2024-03073

“FGF7 as an essential mediator for the onset of ankylosing enthesitis related to psoriatic dermatitis” by Ebihara et al.

Reviewer #1 (Comments to the Authors (Required)):

This is an interesting and straightforward study showing a link between IL-17A and FGF7 expression in the joint. However, the 'emphasis / suggestion' of the authors that 'IL-17A in PsA dermatitis induces the elevation of FGF7 levels in joint enthesitis' is not the whole story as IL-17A can also be produced directly in the joint. The authors do not prove that IL-17A is coming from the skin and that the skin is the (only) source of IL-17A to upregulate FGF7 in the joint causing ankylosing enthesitis. Therefore, this 'message/suggestion' in the manuscript should be rewritten.

Page 19, line 295–297

In fact, we have not directly shown that the only source of IL-17A is from the skin. Since the DBA1/J mice used in this study developed arthritis after the onset of dermatitis due to aggressive behavior, we believe that there is the pathological connection between the skin and joints. Furthermore, IL-17A is involved in that pathological connection as the major parameter, as described previously (Ebihara et al. *Autoimmunity*. 2015;48:259–266). However, IL-17A from other organs may be involved in the pathological connection between the skin and joints, leaving room to consider sources of IL-17A other than skin. Therefore, we described the comments in the discussion section.

Reviewer #2 (Comments to the Authors (Required)):

Major:

-in all figure legends, please indicate the number of experimental replicates, as it is unclear how many times each experiment was performed (3–4 tissues, but from how many separate experiments/batches/mice?). A minimum of two to three repeat experiments from separate biological donors is customary in the field.

Page 50, line 779–780, Page 51, line 781–782, Page 52, line 802–803

In the histoculture data, each dot in Fig1 and 2 represents one experiment with pooled entheses from 3–4 mice. Therefore, we changed the sentence in the figure legend.

-Suggest making a summary figure demonstrating the pathway and joint-specific findings

Page 20, line 303, Page 55, line 859–863, Figure 6

We made a summary figure demonstrating the pathological connection between the skin and joints, and uploaded it as Figure 6. In addition, we added the sentence in the figure legends section, and the Figure number (Figure 6) in the discussion section.

Minor:

-lines 47-48 p5: suggest rewording to "The prevalence of PsA in the psoriasis population varies widely, most recently estimated to range from 5 to 40%"

Page 5, line 47–48

We changed the sentence in the introduction session, as you suggested.

-line 156 p 11, maybe consider mentioning koebnerization here?

Page 11, line 158–160, Page 46, line 710–712

We added the sentence and a reference on the Koebner phenomenon.

-line 287 p 19 missing the i for in

Page 19, line 288

We corrected the mistake.

-line 302 missing space between synovium of

Page 20, line 309

We corrected the mistake.

Reviewer #3 (Comments to the Authors (Required))

1 FGF7/KGF Binding Specificity

FGF7/KGF is known to bind not only FGFR2IIIb but also FGFR1IIIb. It would be important to perform the same experiments using an FGFR1 antibody to confirm whether FGF7 and IL-6 levels decrease as hypothesized. Testing with SU5402, an FGFR1 inhibitor, may also provide further insights.

Page 13, line 184–185, Page 26, line 412–413

FGF7 is known to bind to FGFR1IIIb, as you commented. However, since anti-FGFR1IIIb-specific antibodies were unavailable, we utilized quantitative RT-PCR instead of FACS analysis to assess FGFR1IIIb expression in the joints of the PsA model. The analysis revealed no detectable expression of FGFR1IIIb. Accordingly, we added this method to the materials and methods section and included the findings in the results section.

Furthermore, we think that the anti-FGFR1 antibodies likely target not only FGFR1IIIb but also FGFR1IIIc. As a result, these antibodies are unsuitable for this study, which aims to identify the pathological effects of specifically suppressing FGF7 signaling.

SU5402 is known to inhibit not only FGFR1 but also PDGFR β and VEGFR (Widberg et al. *Am J Physiol Endocrinol Metab.* 2009;296:E121–31). Although it is not suitable for the aim of this study due to specificity issues, we believe that it has potential as a therapeutic agent for PsA.

2 Alternative Inhibitors for FGFR2IIIb

Instead of receptor-blocking antibodies like FGFR2IIIb, it would be valuable to evaluate the efficacy of small molecules, such as TAS-120 (Futibatinib), or ligand traps (if available) for FGFR2IIIb. If similar results are obtained, it would support the utility of these small molecules in therapeutic applications.

TAS-120 (fucasatinib) has been reported to be FGFR1–4 inhibitor (Sootome et al., *Cancer Res.* 2020;80:4986–4997). Since the aim of this study was to specifically inhibit acquired FGF7 signaling with a highly selective antibody, we

think that TAS-120 is not suitable for use in this study. However, we believe that it has potential as a therapeutic agent for PsA. And also, we have no idea on how exactly to trap FGF7 for FGFR2IIIb.

January 29, 2025

RE: Life Science Alliance Manuscript #LSA-2024-03073R

Dr. Shin Ebihara
Tohoku University Graduate School of Medicine
Department of Pathology
2-1 Seiryō, Aoba
Sendai, Miyagi 980-8575
Japan

Dear Dr. Ebihara,

Thank you for submitting your revised manuscript entitled "FGF7 as an essential mediator for the onset of ankylosing enthesitis related to psoriatic dermatitis". We would be happy to publish your paper in Life Science Alliance pending final revisions necessary to meet our formatting guidelines.

- please be sure that the authorship listing and order is correct
- please add the Twitter/X and Bluesky handles of your host institute/organization as well as your own or/and one of the authors in our system
- please add callouts for Figure S1A-I to your main manuscript text
- you may want to consider uploading Figure 6 as a Graphical Abstract rather than as a figure, but this is up to you

A. FINAL FILES:

B. MANUSCRIPT ORGANIZATION AND FORMATTING:

Thank you for your attention to these final processing requirements. Please revise and format the manuscript and upload materials within 5 days.

Sincerely,

January 31, 2025

RE: Life Science Alliance Manuscript #LSA-2024-03073RR

Dr. Shin Ebihara
Tohoku University Graduate School of Medicine
Department of Pathology
2-1 Seiryō, Aoba
Sendai, Miyagi 980-8575
Japan

Dear Dr. Ebihara,

Thank you for submitting your Research Article entitled "FGF7 as an essential mediator for the onset of ankylosing enthesitis related to psoriatic dermatitis". It is a pleasure to let you know that your manuscript is now accepted for publication in Life Science Alliance. Congratulations on this interesting work.

DISTRIBUTION OF MATERIALS:

Again, congratulations on a very nice paper. I hope you found the review process to be constructive and are pleased with how the manuscript was handled editorially. We look forward to future exciting submissions from your lab.

Sincerely,
